# Phenotype-Driven Diagnostic of *PTEN* Hamartoma Tumor Syndrome: Macrocephaly, But Neither Height nor Weight Development, Is the Important Trait in Children

**DOI:** 10.3390/cancers11070975

**Published:** 2019-07-11

**Authors:** Michaela Plamper, Bettina Gohlke, Felix Schreiner, Joachim Woelfle

**Affiliations:** Pediatric Endocrinology and Diabetology Division, Children’s Hospital, University of Bonn, Adenauerallee 119, 53113 Bonn, Germany

**Keywords:** PTEN, head circumference, PHTS, children

## Abstract

*PTEN* hamartoma tumor syndrome (PHTS) encompasses different syndromic disorders which are associated with autosomal-dominant mutations of the tumor suppressor gene *PTEN*. Patients are at high risk to develop benign and malignant tumors. Macrocephaly is a diagnostic feature, but there is a paucity of data on auxological development during childhood. Growth charts for height, weight and head circumference for PHTS do not exist yet. In this study, patient data for height, weight and head circumferences (HC) were collected from repeated medical exams or prevention check-up visits starting at birth. Growth charts were generated and compared to German reference data. Standard deviation scores (SDS) of HC, height and body mass index (BMI) were calculated. We included 23 pediatric patients (8 female, 15 male) with molecular proven *PTEN* gene mutation. Most male patients already demonstrated macrocephaly at birth (73%), whereas only one female patient had documented congenital macrocephaly. By the age of two years all patients exhibited a head circumference above the 97th percentile. Stratified for different age groups the median HC-SDSs were between +3.3 and +5.5 in male patients and between +2.9 and +4.1 in female patients. Height, weight and BMI measurements for both sexes were mostly within the normal range. We conclude that macrocephaly, but not height, weight or BMI, is useful in the identification of PHTS patients. The increased HC in PHTS patients develops early in life and is more pronounced in males than in females, which might explain the finding of a higher percentage of male PHTS patients diagnosed during childhood.

## 1. Introduction

*PTEN* hamartoma tumor syndrome (PHTS) encompasses different syndromic disorders (Cowden syndrome, Bannayan Riley Ruvalcaba syndrome, Lhermitte Duclos syndrome, Proteus and Proteus-like syndrome, juvenile polyposis of infancy, autism spectrum disorders with macrocephaly or children with macrocephaly and developmental delay/cognitive impairment), which are all associated with germline autosomal-dominant mutations of the tumor suppressor gene *PTEN*.

Bannayan-Riley-Ruvalcaba syndrome (BRRS) and Cowden syndrome (CS) can be seen as one condition with variable expression and age-related penetrance [1,2]. Patients with Cowden syndrome (CS) especially are at a significantly increased risk for the development of benign and malignant tumors. The most frequently affected organs are the breast, endometrium and thyroid, but colorectal, renal, central nervous system, and skin tumors have also been reported [3]. It is noteworthy however, that in addition to CS and BRRS, all individuals with PHTS have an increased risk of malignancy and benefit from cancer surveillance strategies.

In the diagnostic workup of subjects with CS [4,5,6] and BRRS [1,7], macrocephaly (MC) is an important diagnostic feature. Whereas several forms of cancer and polyps are common in adult patients, macrocephaly with or without neurodevelopmental disorders is frequently the only or the most evident symptom in childhood, thereby prompting a diagnostic workup. However, to date there is limited data available on prevalence, degree and longitudinal development of head circumference and other auxological parameters during childhood [8,9,10]. Syndrome-specific longitudinal data and growth charts for length/height, weight and head circumference do not exist yet.

## 2. Methods

We conducted a single-centered study on children and adolescents with molecularly proven *PTEN* hamartoma tumor syndrome, who presented in our pediatric endocrine clinic between the years 2013 and 2019. We included 23 patients (15 male, 8 female) with a current age range between two and 17 years (age range in boys from two to 17 years of age; in girls from three to ten years of age).

All available auxological data (length/height, weight and head circumference) were collected from medical reports and from the standardized German prevention check-up exams. Data were collected retrospectively starting at patient’s birth and prospectively from the first visit in our clinic until 2019. The collection of clinical data and cerebral magnetic resonance imaging (MRI) was performed prior to a diagnosis of PHTS or after confirmation of PHTS as part of a symptom-related workup. Growth charts were generated for height, weight and head circumference (HC) of each patient and compared to German reference data [11]. Standard deviation scores (SDS) for head circumference, height and BMI were calculated. Familial target height was calculated according to Tanner [12].

Macrocephaly was defined as a head circumference above the 97th percentile of German reference data from the KiGGS surveillance program [11]. Overweight was defined as a BMI above the 90th percentile, and obesity as a BMI above the 97th percentile [11].

Standard deviation scores (z-scores) were calculated by using the LMS method. It was chosen as it summarizes the data in terms of three age-specific curves called lambda (L), mu (M) and sigma (S) base on German population- specific data [13]. The calculation is based on the following formula:
Z = ((X/M)^L^ − 1)/(S × L), for L not 0.

Because of the relatively small number of cases at different timepoints, patient data in growth charts are displayed as complete years. Patient’s age was defined as X years for patients between X ± 0.5 year (e.g., a patient’s age would be defined as 5 years for children aged 4.5 to 5.4 years).

## 3. Results

We included longitudinal auxological data from a total of 23 patients (8 female and 15 male patients) with molecularly proven *PTEN* gene mutation.

### 3.1. Head Circumference

Birth data for all male patients were available. The majority of male patients exhibited macrocephaly already at birth (11/15, corresponding to 73%). Even though two patients with a head circumference below the 97th percentile were born preterm (34 + 5 and 35 + 0 weeks of gestation), there was still no difference in the percentage of macrocephaly when corrected for gestational age.

At the age of two years all male patients exhibited a head circumference above the 97th percentile (Figure 1).

Birth data from our female patients were available for 7/8 patients and corrected for gestational age. Only one female patient had documented congenital macrocephaly (14%), whereas all other female patients presented a head circumference between the 75th and 97th percentile. However, at the age of 0.5 years, HC was above the 97th percentile in all patients with data available (Figure 2).

12 out of 22 patients (55%) were born with macrocephaly. The method of delivery could be elicited for 20 patients. 14 of those 20 patients were born by Caesarean section (70%).

Stratified for different age groups, the median HC-SDSs of boys were between +3.3 and +5.5 (mean 4.2 SDS) (Appendix A). Median HC-SDS was lowest at the age of six and seven years and highest at the age of eleven. Outliers can be explained by the paucity of data for these age groups, with measurements of HC being available only in two patients (at seven years) respectively three patients (at six and eleven years).

Female median HC-SDSs were between 2.9 and 4.1 (mean 3.3) (Appendix A).

In our study, mean and median HC-SDS of male patients was almost one SDS higher than those of girls with PHTS. This finding was consistently present in all age groups (except for the few outliners described before).

### 3.2. Length/Height

Height development of boys and girls was within the normal range for most time points. All female patients grew within the adequate range for their individually calculated target height. Three out of the 15 boys (20%) demonstrated a growth pattern with a height above their respective familial target height range. All remaining boys (80%) exhibited a growth pattern corresponding to their parent’s heights (Figure 3 and Figure 4).

### 3.3. Weight and BMI Development

The majority of patients had a BMI-SDS above the 50th percentile (Appendix A). Especially in the younger age groups, the percentage of overweight and obese patients was higher compared to German reference data (Table 1).

### 3.4. Additional Brain MRI and Clinical Characteristics of the Cohort

#### 3.4.1. Brain MRI Scans

15 of the 23 patients underwent a cerebral MRI scan. In all cases, the first MRI scan was performed within the diagnostic work-up of macrocephaly. 13 of 15 patients exhibited noticeable variations in the MRI; seven patients presented with white matter abnormalities. In three patients, enlarged perivascular spaces were found (Virchow Robin spaces). Additional pathologies included a Chiari malformation type I, arachnoid cysts, focal cortical dysplasia, a cavernoma and a clinical diagnosis of pseudotumor cerebri in one patient, respectively (Table 2).

#### 3.4.2. Clinical Features

14 of the 23 (61%) patients exhibited thyroid pathology, whereas nine patients had no pathologies detected so far with ultrasound screening. One third of those without pathologies were less than three years, one third were between three and ten years of age, the last third were aged between eleven and thirteen years. Of those patients exhibiting pathology with ultrasound screening, seven patients (50%) underwent thyroid surgery due to highly suspicious lesions (age range six to thirteen years). Histopathology included nodular goiter, follicular adenoma, papillary microcarcinoma in a six-year-old-boy and follicular carcinoma in a 13-year-old girl. Additional pathologies comprised autoimmune thyroid disease, thyroid nodules, which are still under regular observation, and thyroid cysts (Table 2). Some of these findings have already been published [14]. Other features in PHTS patients of this cohort were lipoma, hemangioma, trichilemmoma and penile freckling. Until now colonoscopy has been performed in three patients. All of them revealed multiple polyps of the gastrointestinal tract. One patient carried an additional *BMPR1A* deletion and exhibited severe gastrointestinal symptoms. He underwent colectomy for disease control. In most cases, parents reported a relatively late motor development or a motoric clumsiness, as well as muscle hypotonia. Developmental delay and autism were documented in some cases (Table 2). Most of these patients fulfilled the criteria for BRRS. However, in several of these patients it was difficult to distinguish between a diagnosis of BRRS and CS, due to the difficulty when dealing with pediatric cases.

## 4. Discussion

In this study macrocephaly was found in all PHTS patients. Macrocephaly is a required criterion for diagnosis of *PTEN* hamartoma syndrome in childhood. A prevalence of 100% has been reported in several publications concerning pediatric patients [15,16,17,18]. This is also consistent with previous studies, which included adult patients with molecularly proven *PTEN* gene mutation, reporting macrocephaly prevalence between of 93% [19] and 100% [2,8,15,20]. One previous study reported a lower frequency of MC (80%) in patients with a clinical diagnosis of CS. However, this study was undertaken before *PTEN* was identified as the predisposing gene of CS [9]. In our study we observed a more pronounced degree of macrocephaly in male children and adolescents (mean HC-SDS + 4.2) compared to female patients (mean HC-SDS + 3.3). This sex discordance for the degree of HC-SDS might explain why more boys than girls presented with a diagnosis of a *PTEN* gene mutation in our institution. In contrast, Mester et al. found no sex difference in the degree of macrocephaly in pediatric patients and described a high percentage of patients with an occipital-frontal circumference (OFC) beyond + 5 standard deviation (SD) in childhood [8]. Kato et al. identified six patients with *PTEN* mutation who showed macrocephaly with a degree of +3.2 to 6.0 SD [17].

Although macrocephaly is a well-known required criterion for *PTEN* testing in children, the natural history of head circumference growth is still largely unknown. Our study reports longitudinal data of children with PHTS, showing that the increase in head circumference occurs early in infancy. Frequently macrocephaly is already present prenatally, which probably explains the high rate of Caesarean sections in our cohort. More male (73%) than female newborns (14%) showed macrocephaly at birth. By the age of two years all children exhibited macrocephaly, even those who were born preterm. Some authors reported that macrocephaly persisted from birth into adulthood [21,22,23]. Others described a progression in the degree of head circumference and macrocephaly. Tan et al. [16] reported a 13% incidence of macrocephaly in newborns, Ciaccio et al. [10] presented two of 16 patients (14%) who exhibited macrocephaly at birth. Busa et al. [18] reported a series of six patients, two of whom had a diagnosis of macrocephaly during pregnancy and four presented macrocephalic at birth.

At a very young age, macrocephaly often seems to be the only noticeable feature of PHTS. Other symptoms, such as skin manifestation (lipoma, penile freckling), thyroid disease, hamartomatous polyps of the gastrointestinal tract or associated tumor diseases usually develop later in life. Considering that MC is a diagnostic feature in several syndromal disorders, diagnosis of PHTS in childhood is especially challenging. Therefore, in our view, which is also reflected in the German national PHTS guideline for children [24], genetic testing for *PTEN* gene mutation should be less restrictive in childhood [16,24] than recommended in the National comprehensive cancer network (NCCN) Guidelines, which addresses adult patients [5]. Based on our findings, we recommend genetic testing in children and adolescents with macrocephaly (defined as occipitofrontal circumference > 2 standard deviations over the populations mean, or 97th percentile) and at least one of the following additional features (modified after Tan), [16]: autism or developmental delay, skin abnormalities (like lipoma, hemangioma, penile freckling or others), vascular malformations, gastrointestinal hamartoma or thyroid disease, in particular adenoma and carcinoma. However, to date there are still few reports on patients with tumor disease and proven *PTEN* mutation [8] who did not demonstrate macrocephaly, which underlines the importance to initiate genetic testing not only in patients with macrocephaly, but also in those who fulfill other major and minor criteria for disease diagnosis.

Early diagnosis is important, for several reasons. First a diagnosis aids in explaining the child’s peculiarity and associated symptoms, such as developmental delay, autism, and muscle weakness [25]. Secondly, a delayed and/or misleading diagnostic work up can be avoided. But most importantly a surveillance program for early tumor detection should be installed immediately after diagnosis due to the increased risk of development of thyroid carcinoma and melanoma in early childhood [3,14,19,20,26].

Pavone et al. divides the occurrence of a head circumference of + 2 SDS into categories of megalencephaly and macrocephaly. Megalencephaly defines an increased growth of cerebral structures related to dysfunctional neuronal proliferaton and/or migration phases. Macrocephaly is linked to various events that can result in an increase of orbito-frontal head circumference. PHTS is assigned to the group with anatomic megalencephaly, which is linked to a single gene mutation. The differential diagnosis of combined megalencephaly and gigantism includes Sotos syndrome, Weaver syndrome and Simpson-Golabi-Behmel syndrome [27]. Besides megalencephaly, several patients in this cohort exhibited variable white matter abnormalities (Table 2). In addition to analyzing frequency and longitudinal development of head circumference in PHTS subjects, we speculated that length and weight development in combination with macrocephaly could improve an early identification of pediatric patients with *PTEN* gene mutation. However, in this study of 23 patients, height development was mostly within the normal range and within the familial target height of the child. Even though the percentage of overweight and obese patients was relatively high in the younger age group, and most patients exhibited a BMI above average (>50th percentile for sex and age), this was not a constant attribute and therefore could not help to distinguish PHTS patients from other children. Thus, our data do not support using an auxological screening for these patients apart from focusing on abnormal head circumference. A limitation of our study is the relatively small number of patients and that most of our patients have not yet reached adulthood. Therefore, it is not possible to build specific growth charts for PHTS at this point in time. To create growth charts for this rare disease, it would be helpful to initiate national and international registers with standardized measurements of head circumference, height and weight during childhood and adolescence.

## 5. Conclusions

Macrocephaly, but no other auxological parameter (length/height, weight, BMI), is helpful in the early identification of PHTS patients. The increase in HC in PHTS patients develops early in life and is more pronounced in males than in females, which might explain the fact that a higher percentage of male PHTS patients are diagnosed during childhood.

## Figures and Tables

**Figure 1 cancers-11-00975-f001:**
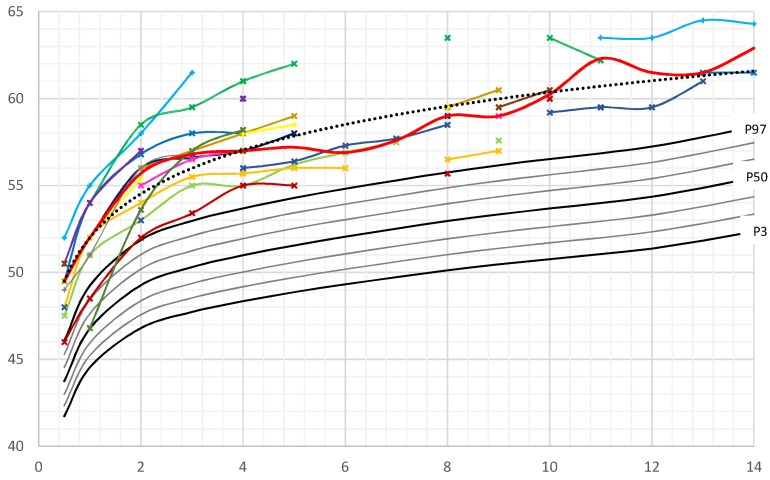
Head circumference (HC) of male *PTEN* hamartoma tumor syndrome (PHTS) patients in comparison to German reference growth charts [11]. Legend: ___ (red): Median, ……: Trendline, ___: P3-P97. Colored lines: individual patient’s HC curves.

**Figure 2 cancers-11-00975-f002:**
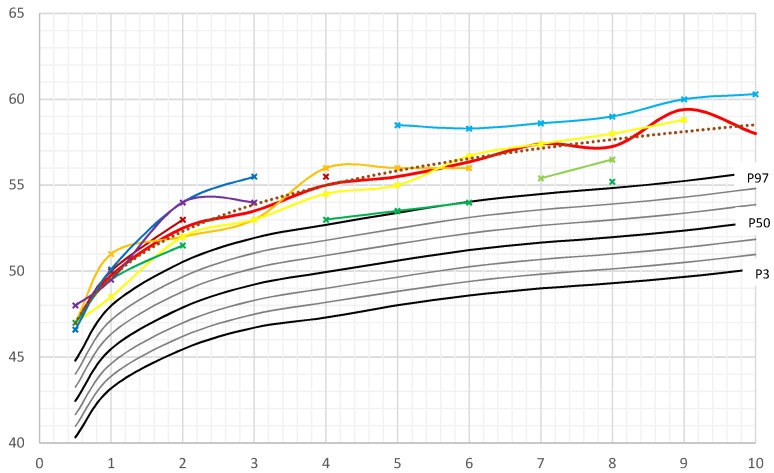
Head circumference (HC) of female PHTS patients in comparison to German reference growth charts [11]. Legend: ___(red): Median, ……: Trendline, ___: P3–P97. Colored lines: individual patient’s HC curves.

**Figure 3 cancers-11-00975-f003:**
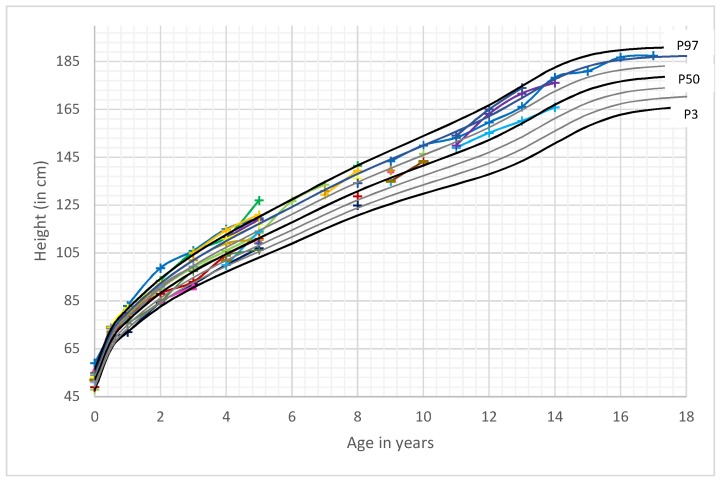
Height development of male PHTS patients in comparison to German reference growth charts [11]. Legend: ___: P3–P97. Colored lines: individual patient’s length curves.

**Figure 4 cancers-11-00975-f004:**
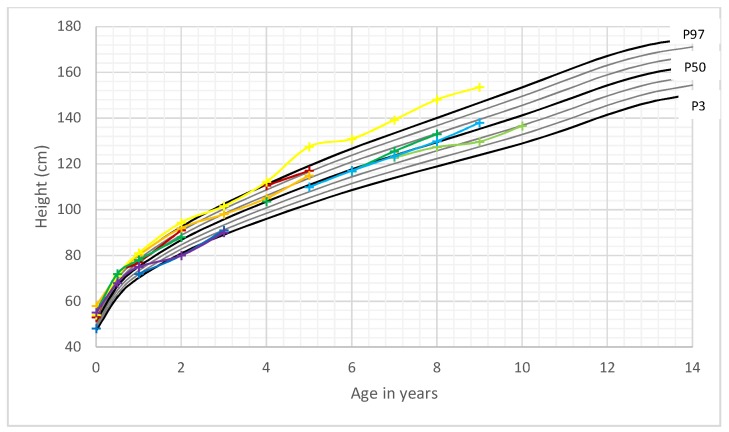
Height of female PHTS patients in comparison to German reference growth charts [11]. Legend: ___: P3–P97. Colored lines: individual patient’s length curves.

**Table 1 cancers-11-00975-t001:** Rate of overweight and obesity in pediatric patients with PHTS.

Age in Years	*n*	Overweight *n* (%)	Obesity *n* (%)
0.5	16	4 (25%)	1 (6.25%)
1	17	2 (11.8%)	2 (11.8%)
2	18	10 (55.5%)	3 (16.7%)
3	13	4 (30.8%)	0
4	16	7 (43.8%)	6 (37.5%)
5	17	4 (23.5%)	4 (23.5%)
6	4	1 (25%)	0
7	8	3 (37.5%)	0
8	9	2 (22.2%)	1 (11.1%)
9	8	1 (12.5%)	0
10	5	2 (40%)	0
11	4	0	0
12	4	0	0
13	4	0	0

**Table 2 cancers-11-00975-t002:** Additional clinical characteristics, magnetic resonance imaging (MRI) features and molecular findings of pediatric PHTS patients. n.d. (not detected)

**Patient No (Male)**	**Mutation/Deletion in *PTEN* Gene (Localisation)**	**MRI Features**	**Add. Clinical Features (Except Macrocephaly)**
1	c.389G>A; Arg130 Gln (exon 5)	n.d.	Follicular adenoma (thyroid), multiple GI polyps, lipoma, hemangioma
2	c.389G>A; Arg130 Gln (exon 5)	Virchow-Robin spaces (enlarged perivascular spaces) white matter abnormalities	Papillary microcarcinoma in follicular adenoma (thyroid) [14], autism, developmental delay, lipoma, penile freckling,
3	c.540C>A; p.Y180X (exon 6)	Virchow-Robin spaces	Nodular goiter, lipoma, penile freckling
4	c.737C>T.p.Pro246Leu (exon 7)	n.d.	Autoimmune thyroid disease, haemangioma, penile freckling
5	c.209+5G>A (Intron 3)	White matter abnormalities	Developmental delay
6	c.445C>T; Gln149X (exon 5)	White matter abnormalities	Autoimmune thyroid disease, penile freckling, trichilemmoma
7	c.509G>A; pSer170Asn (exon 6)	n.d.	Penile freckling
8	heterozygous deletion (exon 1–2)	Pseudotumor cerebri, Ventriculoperitoneal shunt	Multiple gastrointestinal polyps, lipoma, hydronephrosis
9	partial deletion (exon 6)	White matter abnormalities	Penile freckling, muscle hypotonia, lipoma
10	c.697C>T; pArg233 *(exon 7)	n.d.	Two follicular adenoma (thyroid), lipoma
11	c.959T>G (p.Leu320 *)	Cavernoma	Thyroid adenoma, penile freckling, lipoma, developmental delay
12	c.987dup T (p.Lys330 *) (exon 8)	n.d.	Colloid cysts of thyroid, lipoma, haemangioma
13	c.(492+1_493-1)_(1026+1_1027-1)del	Focal cortical dysplasias	Moderate developmental delay, lipoma
14	heterozygous deletion *PTEN* and *BMPR1A* Gene	Arachnoid cysts left and right of the pineal region	Little lesions of left thyroidal lobe. Additional *BMPR1A* deletion, severe gastrointestinal disease, juvenile polyposis requiring colectomy, penile freckling, dilatative cardiomyopathy
15	c.800_801delAG (exon 7)	Enlarged periventricular spaces, Virchow-Robin spaces	Moderately reduced IQ, scoliosis,
**Patient No. (Female)**	**Mutation in *PTEN* Gene (Localisation)**	**MRI Features**	**Add. Clinical Features (Except MC)**
1	c.741dupA; p.Pro248Thrfs*5 (exon 7)	n.d.	Follicular carcinoma and microfollicular adenoma of thyroid, lipoma
2	c.302T>C; p.Ile101Thr (exon 5)	n.d.	Suspicious lesion in ultrasound screening of thyroid, developmental delay, trichilemmoma
3	c.762dupA;p.Val255Serfs*43 (exon 7)	White matter abnormalities	Lipoma, developmental delay, precocious puberty
4	c.49C>T;p.Gln17* (exon1)	Normal MRI Scan	Nodular goiter, lipoma
5	c.1008C>G;p.Tyr336* (exon 8)	n.d.	Suspicious lesion (hyper-perfusion, microcalcific.) of the thyroid, lipoma, developmental delay
6	c.492delG;p.Gly165Glufs*2 (exon 5)	Normal MRI Scan	Follicular nodule (thyroid), lipoma
7	c.1133_1136del.pArg378ilefs*37 (exon 9)	Dysmyelinisation, microgyria, Chiari malformation I	Developmental delay, muscle hypotonia, diarrhea
8	c.389G>A; p.(Arg130 Gln) (exon 5)	White matter abnormalities	Family history, muscle hypotonia, cutis laxa, developmental delay

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
