# Peer review of "Phenotype-Driven Diagnostic of PTEN Hamartoma Tumor Syndrome: Macrocephaly, But Neither Height nor Weight Development, Is the Important Trait in Children"

_cancers, 2019, doi:10.3390/cancers11070975_

Reviewer 1 Report

Macrocephaly is a a major diagnostic criteria for PHTS together with breast, thyroid and endometrium cancer, benign hamartomas and skin lesions. 

The authors in this article wanted to seek whether measurements, such as height and weight, taken during the child development, could be used as diagnostic criteria for PHTS together with macrocephaly and whether any correlation was present. 

The data collected here have proven that, as for now among the three, macrocephaly is the only feature that can be used as diagnostic criteria. 

-It would be good if the PTEN mutations identified in these patients were listed and also the type of syndromic disorder they have been associated with. 

It would be better if the measurements were shown in a more clear manner. It is hard to interpret the graphs. Are symbols across graphs assigned always to the same patient?

Author Response

The referee asked for more genetic and phenotypical information in our patients. We therefore included a new section (line 127 ff.) “Additional clinical characteristics of our cohort” and added a table (table 2), listing the mutations, MRI results and other clinical features of our patients. In most of our pediatric patients the phenotype was in congruence with BRRS syndrome. However, at this young age it remains difficult to distinguish BRRS and CS, since a CS phenotyp might evolve in subjects who were formerly classified as BRRS.

In addition, ref. 1 recommended to show the measurements in our figures more clearly. We tried to differentiate the different patients more clearly by choosing different colors and marked the single measurements with a cross in the same color.

Reviewer 2 Report

The manuscript submitted by Plamper et.al “Phenotype-driven diagnostic of PTEN hamartoma Tumor Syndrome: Macrocephaly, but neither height nor weight development, is the important trait in children” is not suitable for the journal in current format. The sample size used for the analysis is very small. The experimental data is not sufficient and is not conclusive.

Author Response

We agree, that our sample size remains relatively small. However, our cohort consisting of 23 patients is still the largest pediatric German cohort and to the best of our knowledge the largest single center pediatric European cohort. We agree that to date it is not possible to build growth charts for PHTS at that point in time. Our goal was to compare the longitudinal auxologic data of this German cohort with reference data in order to test whether head circumference, height and weight might be useful phenotypical parameters in an early diagnosis of PHTS. To create growth charts for this rare disease, it would be helpful to initiate national and international register with standardized measurements of head circumference, height and weight during childhood and adolescents. Our paper focusing on macrocephaly as an important phenotypical feature might aid in an earlier and more frequent identification of PHTS patients.

Reviewer 3 Report

The authors present a very interesting series of pediatric patients with PTEN mutations, who are followed up for their auxological development from birth. The study contains the first specific growth curves for the PTHS.

There are no major problems with the design of the study and with the presentation of the data. However, here are some issues and suggestions:

- Mat & Met: the authors should introduce some ethic consideration about their patients’ data handling

- Discussion:

1) the authors recommend testing children with macrocephaly associated with other features, which are part of the PTHS spectrum, but they present no data in support of those latter. Do they have data about?  Or, following the evidences of their study, restrict their recommendation to the auxological features and in this regard, if they feel so, they could be more assertive: is the macrocephaly alone a criterion for testing children? Above what threshold? Could be normal stature an additional feature? The authors could discuss more thoroughly such points in respect of the results they had from their study

2) They present the subdivision by Pavone et al., but they do not comment about that: are their data in agreement? Do they have MRI data about their patients?  

3) one small caveat about the number of patients, which is still small to build growth charts should be added

Minor points:

Behmel, not Behmal

Tan et al., not TAN

Author Response

You recommended that we should introduce some ethical considerations about our patients’ data handling.

However, all data presented in this report were part of standard routine care. Therefore, we did not apply for an ethical vote.

Discussion Point 1: As we recommended in our discussion to test children with macrocephaly and additional other features for PTEN mutation, you asked if we have data to support this statement. Furthermore, you asked if a normal stature could be an additional feature? And above what threshold of MC patients should be tested.

We added a new section and table 2, concerning additional clinical features to describe additional clinical features of our patients.

We would recommend patients to be tested for PTEN mutation if these exhibit an occipitofrontal circumference > 2 standard deviations over the populations mean, or 97th percentile and the mentioned additional features.

To date, our database is not sufficient to calculate specific sensitivity and specificity of this threshold and has to be reevaluated once more data in children and adolescents are available. From our perspective it is too early to decide whether the combination of MC and normal stature could be an additional feature of PHTS patients.

Discussion Point 2: R: They present the subdivision by Pavone et al., but they do not comment about that: are their data in agreement? Do they have MRI data about their patients?  

Answer: New created table 2 gives information about MRI data of our patients as well as the underlying mutation and additional clinical features. We added the following sentence: “In accordance to that in our cohort several patients showed variations of the white matter (table 2).” (line 213)

Point 3:

We included the following information: “A limitation of our study is the small number of patients and that most of our patients are not full-grown up yet. Therefore, it is not possible to build growth charts for PHTS at that point in time.“ (line 223 ff.).

We corrected the minor points mentioned by the referee.

Reviewer 4 Report

In this paper Plamper and coauthors report the auxological data of 23 pediatric patients with a germline heterozygous PTEN mutation. Data about Head Circumference, Height and BMI have been collected both retrospectively and prospectively, and then compared with German reference growth charts with the purpose to describe the growth pattern of children with PHTS. The ultimate goals are to construct disease-specific growth charts which do not yet exist, and to provide clinical data useful in the early identification of PHTS patients. In conclusion, the authors show that macrocephaly, but not height, weight or BMI are useful data to identify PHTS patients; in addition they describe the natural history of increased head circumference, more pronounced in males than in females. 

PTEN Hamartoma Tumor syndrome (PHTS) is an umbrella term used to comprise a spectrum of autosomal dominant disorders associated with a germline PTEN mutation. Pathogenic heterozygous germline PTEN mutations are implicated in familial cancer predisposition, such as Cowden and Bannayan-Riley-Ruvalcaba syndrome; in Proteus and Proteus-like syndrome, malformation syndromes with segmental overgrowth associated or not with intellectual disability; and in children with macrocephaly and autism or developmental delay/cognitive impairment.

An overarching question is to ascertain the range of phenotypes and clinical presentation in children with pathogenic PTEN mutations to help the development of diagnostic criteria as well as evidence-based indications for PTEN genetic testing. It is important to identify appropriate clinical criteria to guide selection for PTEN testing since all individuals with PHTS have an increased risk of malignancy and benefit from cancer surveillance strategies.

Distinct adult and pediatric diagnostic criteria for PTEN mutation testing are of considerable clinical utility. Macrocephaly and neurodevelopmental disorders are well-established presenting clinical features in children, whereas cancers and polyps are common presenting symptoms in adults.

Early pediatric diagnosis is crucial for medical and developmental surveillance as well as for testing other at-risk family members.

Macrocephaly (occipitofrontal circumference > 2 standard deviation over the population mean, or 97.5 th percentile) is a well-known characteristic trait affecting all PHTS children. It represents a necessary criterion for diagnosis, based on 100% prevalence at the time of diagnosis for PTEN mutated children in several papers (Tan et al., 2007; Tan et al., 2011; Kato et al., 2018; Busa et al., 2015). In the series described by Hansen-Kiss and coauthors (2017), 1 out of 47 children had only relative macrocephaly at the time of diagnosis; the brain MRI of the child however, revealed bilateral asymmetric areas of tissue injury and volume loss in the periventricular white matter of the frontal lobes, consistent with a possible prenatal stroke.

If macrocephaly is a well-known necessary criterion for PTEN testing in children, natural history of head circumference growth are still unclear and has not been well documented.

Some authors reported  macrocephaly presenting from birth and persisting into adulthood (Piccione et al., 2013, Diliberti et al., 1998, Hansen-Kiss et al 2017 ), while some others described progressive macrocephaly. Ciaccio et al reported only 2 patients among 16 (14%) with a documented macrocephaly at birth, corrected for gestational age. This percentage is almost the same of that of 13% reported by TAN et al., 2007. In the series of Busa and coauthors, macrocephaly was diagnosed during pregnancy in two patients and presented at birth in 4 patients out 6. 

The aim of the study by Plamper to describe auxological growth pattern is therefore interesting, but there are unfortunately some limitations so this work does not add much to the knowledge and the diagnostic work-up of the disease.

First of all, data are too limited, due to the small number of patients, the  small amount of measurements for some patients and the differences in the timepoints of collection between patients. Constructing specific growth charts for PTHS requires detailed and standardized measurements and a larger study sample size. Moreover, head circumference at birth should be corrected for gestational age. In addition, in my opinion, data about previous studies reporting the same information about head growth pattern and gender differences, are not well described in the discussion. See all the previous studies reported in this comment for head growth pattern and the study by Hansen-Kiss (2017) who reported an average head circumference of +4.9 SD in 28 males and +5.58 SD in 18 female patients.

There are some imprecisions in text, such as ZNS in line 44.

Line 92: 12/22 not 12/23 as in one female patient the head circumference at birth is not available.

Line 145-149: the authors don’t report the associated clinical features of their cohort; this part is therefore not pertinent.

Author Response

We would like to thank you for your very constructive review. We added the missing references you suggested in your review and gave more detailed information about previous studies reporting on head circumference and gender differences in the discussion (line 157 ff., line 171 ff.).

Furthermore, we corrected head circumference at birth for gestational age and included this information in the manuscript (line 85-89). However, after correction for gestational age there was no difference in the percentage of macrocephaly.

We agree, that our sample size remains relatively small. However, our cohort consisting of 23 patients is still the largest pediatric German cohort and to the best of our knowledge the largest single center pediatric European cohort. We agree that, it is not possible to build growth charts for PHTS at that point in time. To create growth charts for this rare disease, it would be helpful to initiate national and international register with standardized measurements of head circumference, height and weight during childhood and adolescents. We included a comment about the limitations of our study in the discussion (line 223 ff.).

You also mentioned the following comments:

“There are some imprecisions in text, such as ZNS in line 44.”

We corrected that.

“Line 92: 12/22 not 12/23 as in one female patient the head circumference at birth is not available.”

We corrected that.

“Line 145-149: the authors don’t report the associated clinical features of their cohort; this part is therefore not pertinent.”

We included a table (table 2) and a section (line 127 ff) about associated clinical features of our cohort. Therefore, we left this part in the discussion.

Round  2

Reviewer 2 Report

The authors has addressed the concern and the manuscript can be accepted in the current form. 

Author Response

Thank you for your positive feedback. 

To improve the manuscript, we corrected some spelling mistakes and the paper was revised by a native English speaker, who rewrote some sentences. We hope, that the language will now suffice the standards of the journal. All changes can be tracked in the corrected version of the manuscript, indication by colour-coded changes.

Reviewer 4 Report

I have read the revised paper by Plamper and coauthors.  

The authors have taken the referees' comments into consideration and have therefore substantially modified the text, which was unacceptable to me before.

I have now changed my opinion and I think that the paper is acceptable after minor revisions.

It is true that the authors do not substantially add anything new to the subject and that patients are relatively few and not all have been followed for a long time; however, the authors are nevertheless the first to have built specific growth charts for PTHS. The design of growth curves is clearer and more useful than the simple description of auxological characteristics of the patients.

Furthermore, the description of the clinical characteristics of a PHTS pediatric series from the point of view of endocrinologists is also useful for other specialists involved in the management of patients.

The text should be corrected by a native English speaker because the discussion is not fluent. In addition there are a lot of mistakes.

Specific corrections follow.

Pag 1 line 21: yet at the end of the sentence (after “exist”)

Pag 1 line 42: germline should be added before “autosomal-dominant”

Pag 2 lines 46-47 should be moved at pag 1 line 42.

Pag 2 line 65: it should be added a phrase about the collection of MRI and clinical data as they are described in the results.

Pag 5, line 126, 127; pag 6 line 134: In my opinion the titles of the added paragraphs, should be:

3.4. Additional brain MRI and clinical characteristics of the cohort

3.4.1  Brain MRI scans

3.4.2 Clinical features

Pag 5 line 132: substitute “an arachnoid cysts” with “an arachnoid cyst” or “arachnoid cysts”

Pag 5 line 130-133 : also cortical dysplasia should be reported in the text

Pag 6 line 148 :  motor development is better than “statomotor development”. What does “motoric unhandiness” mean? Do the authors want to indicate “motor clumsiness”?

Pag 7 line 157: a comma should be added after “childhood”

Pag 7 line 165: the comma after “explain” should be removed

Pag 7 line 185: the comma after “considering” should be removed

Pag 7 line 188: the last sentence is incorrect: it should say “mutation should be initiated less restrictively” or, better, “mutation should be less restrictive”

Pag 8 line 197: substitute “fulfil” with “fulfill”

Pag 8 line 202: substitute “important” with “importantly” and add a comma after

Pag 8 line 203: substitute “risk for the development” with “risk of development”

Pag 8 line 209: substitute “which are” with “which is”

Pag 8 line 209-212: it is not clear the meaning of the sentences’ sequence

Table 2: there are some mistakes:

Substitute “Virow Robin spaces” with  “Vircow-Robin spaces”

Substitute “Perventricular” with  “Periventricular”

Author Response

We corrected the mentioned mistakes in the text and in table 2. The paper was revised by a native English speaker, who rewrote some sentences. We hope, that the language will now suffice your standards. All changes can be tracked in the corrected version of the manuscript, indication by colour-coded changes.